# Addressing the huge poor–rich gap of inequalities in accessing safe childbirth care: A first step to achieving universal maternal health coverage in Tanzania

**Deogratius Bintabara** [ID] *

Department of Community Medicine, School of Medicine and Dentistry, The University of Dodoma, Dodoma, Tanzania

\* bintabaradeo@gmail.com

## Abstract

### Introduction

Despite skilled attendance during childbirth has been linked with the reduction of maternal deaths, equality in accessing this safe childbirth care is highly needed to achieving universal maternal health coverage. However, little information is available regarding the extent of inequalities in accessing safe childbirth care in Tanzania. This study was performed to assess the current extent, trend, and potential contributors of poor-rich inequalities in accessing safe childbirth care among women in Tanzania.

### Methods

This study used data from 2004, 2010, and 2016 Tanzania Demographic Health Surveys. The two maternal health services 1) institutional delivery and 2) skilled birth attendance was used to measures access to safe childbirth care. The inequalities were assessed by using concentration curves and concentration indices. The decomposition analysis was computed to identify the potential contributors to the inequalities in accessing safe childbirth care.

### Results

A total of 8725, 8176, and 10052 women between 15 and 49 years old from 2004, 2010, and 2016 surveys respectively were included in the study. There is an average gap (>50%) between the poorest and richest in accessing safe childbirth care during the study period. The concentration curves were below the line of inequality which means women from rich households have higher access to the institutional delivery and skilled birth attendance inequalities in accessing institutional delivery and skilled birth attendance. These were also, confirmed with their respective positive concentration indices. The decomposition analysis was able to unveil that household's wealth status, place of residence, and maternal educa-tion as the major contributors to the persistent inequalities in accessing safe childbirth care.

**Data Availability Statement:** This study was based on the analysis of existing datasets in the Demographic and Health Survey Program repository that are freely available online with all

identifier information removed (https://dhsprogram.com/data/available-datasets.cfm). The specific names of the datasets are Tanzania DHS 2004-5, Tanzania DHS 2010, and Tanzania DHS 2015-16.

**Funding:** The author received no specific funding for this work.

**Competing interests:** The author has declared that no competing interests exist.

## Conclusion

The calls for an integrated policy approach which includes fiscal policies, social protection, labor market, and employment policies need to improve education and wealth status for women from poor households. This might be the first step toward achieving universal maternal health coverage.

## Introduction

Although pregnancy and childbirth are normal processes they can end in complications that could lead to death, particularly in absence of appropriate interventions to deal with life-threatening conditions at the right time [1]. Globally, around a thousand women die every day during pregnancy, delivery, or shortly thereafter [2]. The majority of these preventable deaths (94%) occur in low and lower-middle-income countries [3, 4]. Compared to other regions, Sub-Saharan Africa (SSA) has a very high number of maternal deaths, accounting for nearly two-thirds (196 000) of global maternal deaths in 2017 [4]. This high number of deaths has raised concern whether the SSA countries can be able to achieve the Sustainable Development Goal (SDG) number 3 by the end of 2030, which addresses major health priorities, including maternal and child health [5]. However, skilled attendance during childbirth has been linked with a reduction in maternal deaths [6]. Therefore, safe childbirth care measured by the proportion of births in health facilities (institutional delivery), as well as the proportion of births with skilled birth attendants has been taken as important maternal healthcare indicators to monitor the progress of safe motherhood outcomes [7–9]. But the recent report showed that a high proportion (approximately 40%) of births in SSA were non-institutional deliveries or attended by skilled health personnel between 2012 and 2017 [10]. In Tanzania, institutional deliveries (63%) and births assisted by skilled health personnel (64%) are still unsatisfactory [11]. Therefore, improving safe childbirth care remains a crucial public health priority in the burden areas such as Tanzania which experience a high maternal mortality ratio (MMR) estimated to be 556 maternal deaths per 100 000 live births [11].

In Tanzania, several social, cultural, and geographical factors have been indicated as major barriers towards accessing safe childbirth care (institutional delivery and skilled birth attendance) [12–14]. Moreover, similar to other low- and middle-income countries [15], poverty (poor–rich inequalities) have been described as a key socioeconomic determinant of women's access to these healthcare services in Tanzania [16]. But it is unclear to what extent, the trend over time, and contributors of poor-rich inequalities in accessing safe childbirth care existing in Tanzania. Therefore, realizing the magnitude, trend, and contributors of inequalities in maternal health services in Tanzania may help to eliminate poor-rich gap inequalities in accessing healthcare as the key barrier toward achieving universal coverage for maternal health services. This prompted the authors to use three surveys (2004, 2010, and 2016) with comparable nationally representative samples of reproductive women from Tanzania, to assess the current extent, trend, and potential contributors of poor-rich inequalities in accessing safe childbirth care (measured by accessing institutional deliveries and skilled birth attendance). The findings of the present study will help to improve the strategies aimed to eliminate existing poor-rich inequalities in accessing maternal health in Tanzania and other countries with a low-resource setting.

## Materials and methods

### Data source

This study used data from 2004, 2010, and 2016 Tanzania Demographic Health Surveys (TDHS) conducted by Tanzania's National Bureau of Statistics (NBS) in collaboration with the Office of the Chief Government Statistician (OCGS), Zanzibar, the Ministry of Health, Community Development, Gender, Elderly and Children (MoHCDGEC), Tanzania Mainland, and the Ministry of Health (MOH), Zanzibar. The technical and financial support for the surveys was provided by ICF International under the DHS program. These surveys have been conducted after every four years to improve the quality of health for Tanzanians [11].

### Study design

The current study analyzed nationwide population-based cross-sectional surveys using information collected by interviewing women between the ages of 15 and 49 years who were either residents or visitors in the household on the night before the survey.

### Study sample and sampling technique

The 2004, 2010, and 2016 TDHS employed two-stage cluster sampling techniques to obtain a sample intended to provide nationally representative results according to all regions of Tanzania. At the first stage, the primary sampling units (PSUs) referred to as clusters (a total of 475 clusters in 2004 and 608 clusters in 2010 and 2016 surveys) were selected from a sampling list consisting of enumeration areas delineated by the 2002 and 2012 Tanzania Population and Housing Census [17]. The second stage involved the selection of households. Before household selection, a complete household listing was carried out for all selected clusters. From the list, 22 households were then systematically selected from each cluster, yielding a representative probability sample of 10,312, 10,300, and 13,376 households for 2004, 2010, and 2016 TDHS.

From the successfully interviewed households, the following total numbers of women have been interviewed: 10,139 women in 2004; 10,329 women in 2010; and 13,266 women in 2016 with an average response rate of 97%. However, the current analysis only included women who had a live birth in the five years preceding the surveys, yielding the final sample of 26953 (8725 women, 8176 women, and 10052 women in 2004, 2010, and 2016 respectively).

### Data collection and processing

The TDHSs used four main types of questionnaires during data collection. However, in this study, the data collected from the Women's Questionnaire was used. After the pretesting of the questionnaires, the finalized and corrected version was used in the main surveys. Data collection was performed by nurses who were trained and qualified to be interviewers through a series of practical tests and examinations. Data entry was performed concurrently with data collection in the field. After the paper questionnaires were completed, edited, and checked by the supervisor, the data were entered into computers or tablets equipped with a data entry program. The double entry process was used during data entry to minimize keying errors. Finally, data editing and cleaning were performed. The details about data collection, processing, and other methodology information can be found elsewhere [11].

### Measurement of variables

**Outcome variables.**   The following two outcome variables were used to assess the safe childbirth care in this study: (1) accessing births in facilities (institutional deliveries), which considered when a woman delivered at any of the following public or private facility; clinic,

dispensary, health center or hospital and (2) accessing skilled birth attendance, which was considered if woman assisted by the following trained health personnel; assistant nurse, nurse/midwife, clinical assistant, clinical officer, assistant medical officer, medical officer, and medical specialist.

**Independent variables.**   The following socioeconomic variables that have been linked both empirically and theoretically with the accessibility of maternal healthcare were selected and included in this study. Maternal age was grouped into "15–19," "20–24," "25–29," "30–34," "35–39," "40–44," and "45–49"; Household wealth status was calculated based on household assets and housing characteristics. During the calculation, households were given scores based on the number and types of consumer goods they owned, ranging from a television to a bicycle or car, plus housing characteristics, such as the source of drinking water, toilet facilities, and flooring material. These scores were derived using a principal component analysis. National wealth quintiles were compiled by assigning the household score to each usual (*de jure*) household member, ranking each person in the household population by their score, and then dividing the distribution into five equal categories ("poorest," "poor," "middle," "rich" and "richest"), each including 20% of the population [11]; The residence was categorized as "Urban" for households located in cities, municipalities and town councils gazetted under the Local Government Act, 1982 [18], and "Rural" for households that were located outside the urban areas. Education level was categorized as "none" for women who had not received any kind of formal education, "primary" for women who completed primary education level, "secondary" for women who completed secondary education level, and "highest" for women who completed college and all university level; Marital status was categorized as "no spouse" for women who were single, divorced, separated or widowed and "living with spouse" for women who were married or living with a partner during the interview; Employment was categorized as "employed" for women who reported to be employed and paid in salary in terms of cash and "not employed" for those who did not have any kind of job and paid in terms of cash. The selection of these variables was based on previous studies related to socioeconomic inequalities in accessing maternal healthcare [19–22].

## Analytical methods

The current study used three methods to assess inequalities in accessing safe childbirth care (measured by accessing institutional delivery and skilled birth attendance). These methods were; (1) the computation of concentration curves, (2) the computation of concentration indices, and (3) the decomposition analysis.

**Concentration curves.**   These were used to assess the patterns of inequalities in accessing institutional delivery as well as skilled birth attendance. Principally, the concentration curves plot the cumulative percentage of the outcome variable (y-axis) against the cumulative percentage of the population ranked by household socioeconomic status (wealth status), beginning with the poorest, and ending with the richest (x-axis). In other words, they plot shares of outcome variable against quintiles of the household wealth status. If everyone, irrespective of her wealth status class, has the same value of safe childbirth care, the concentration curves will be a 45-degree (diagonal) line, starting from the bottom left-hand corner to the top right-hand corner. This diagonal line is known as the line of equality. If, by contrast, the outcome variables (accessing institutional delivery and skilled birth attendance) take higher (lower) value among poorer women, the concentration curve will lie above (below) the line of equality. For example, the further the curve is below the line of equality, the more concentrated the outcome variable is among the rich.

**Concentration indices.** Compared to concentration curves, the concentration indices have the additional advantage of quantifying the degree of socioeconomic-related inequalities in healthcare variables such as accessing institutional delivery and skilled birth attendance. This concentration index is defined as "twice the area between concentration curve and the line of equality." It takes the values bounded between -1 and +1. If the index takes the value of "0" indicates that there is no socio-economic-related inequality. But it takes a positive value when the curve lies below the line of equality, indicating the disproportionate concentration of health variable among the rich, and a negative value when it lies below the line of equality. If the health variable is "good" such as accessing institutional delivery and/or skilled birth attendance, the index with a positive value indicating that the health variable is more among the rich. For computation, the concentration index "$C$" can be calculated by using the following formula (1).

$$C = \frac{2}{\mu} cov(y, r) \tag{1}$$

Where $\mu$ refers to the mean of our outcome variables (accessing institutional delivery and skilled birth attendance), $y$ stands for the values of institutional delivery and/or skilled birth attendance for each participant, $r$ refers to the socioeconomic rank of the household in the wealth status and $cov$ is covariance.

**Decomposition analysis.** The decomposition of concentration indices as previously described by Wangstaff, van Doorslaer and Watanabe [23], was performed to estimate the contribution of each explanatory variable to the inequalities in accessing institutional delivery and skilled birth attendance. They demonstrate that the contributions of each variable to the socio-economic-related health inequality is the product of the sensitivity of heath concerning that variable and the degree of socioeconomic-related inequality in that variable [23]. Therefore, the overall concentration index for the predicted outcome variable is the result of the summation of the contributions made by all explanatory variables under exploration. This can be shown mathematically by using the following formula:

$$C = \sum_{k}(\beta_k \bar{x}_k / \mu)C_k + GC_\varepsilon / \mu \tag{2}$$

Where $C$ is the concentration index, k is variables, $x_k$ is the mean of $x_k$, $C_k$ is the concentration index for $x_k$, $\mu$ is the mean of the health outcome, and $GC$ is the generalized concentration index for the error term. The $(\beta_k X_k)/_\mu C_k$ component in the formula represents the explained part of the concentration index of the dependent variable, and the $GC\varepsilon/\mu$ represents the residual component.

Stata V.16 (StataCorp, College Station, TX) was used for analysis in the present study. For all analyses, the Stata survey set commands were used to adjust for the variability of clustering, and all the estimates were weighted to correct for non-responses and disproportionate sampling.

## Ethical considerations

The current study was based on secondary analysis of existing public domain survey datasets that are freely available online with all identifier information detached. The original TDHSs were reviewed by the Institution Review Board (IRB) of ICF Macro at Calverton in the USA and by the National Institute of Medical Research (NIMR) IRB in Tanzania. Both IRBs ensured that the surveys complied with the laws and norms of Tanzania. The study participants were adequately informed about all relevant aspects of the survey, including its objective

**Table 1. Percentage distribution of women between the ages of 15 and 49 years by selected background characteristics in Tanzania DHS, 2004–2016.**

| Variable | 2004 n = 8725 (%) | 2010 n = 8176 (%) | 2016 n = 10052 (%) |
|---|---|---|---|
| **Maternal age** | | | |
| 15–19 | 5.51 | 5.25 | 6.84 |
| 20–24 | 26.81 | 25.18 | 24.34 |
| 25–29 | 27.45 | 26.20 | 25.15 |
| 30–34 | 21.05 | 18.81 | 18.88 |
| 35–39 | 11.61 | 15.94 | 14.67 |
| 40–44 | 5.69 | 6.70 | 7.92 |
| 45–49 | 1.89 | 1.93 | 2.20 |
| **Wealth status** | | | |
| Poorest | 22.63 | 21.14 | 24.14 |
| Poor | 21.28 | 23.59 | 21.24 |
| Middle | 21.39 | 22.46 | 19.19 |
| Rich | 19.26 | 18.66 | 18.78 |
| Richest | 15.44 | 14.15 | 16.66 |
| **Residence** | | | |
| Rural | 80.62 | 79.70 | 72.87 |
| Urban | 19.38 | 20.30 | 27.13 |
| **Maternal education** | | | |
| None | 26.57 | 25.57 | 20.93 |
| Primary | 69.00 | 68.12 | 64.83 |
| Secondary | 3.31 | 6.04 | 13.37 |
| Higher | 1.12 | 0.28 | 0.88 |
| **Marital status** | | | |
| No spouse | 13.00 | 15.50 | 17.22 |
| Living with a spouse | 87.00 | 84.50 | 82.78 |
| **Employment** | | | |
| Not employed | 77.50 | 65.19 | 55.24 |
| Employed | 22.50 | 34.81 | 44.76 |

and interview procedures. All study participants who accepted participating in the study signed informed consent before the interviews.

## Results

### Respondent's characteristics

As presented in Table 1, a total of 8725, 8176, and 10052 women between 15 and 49 years old from 2004, 2010, and 2016 surveys respectively were included in the analysis. The results show that there were changes in many demographic and socioeconomic profile of the study population from 2004 to 2016. The proportion of women without any kind of formal education decreased from 26.6% in 2004 to 20.9% in 2016. A corresponding decrease was also observed in the proportion of women who were not employed from 77.5% in 2004 to 55.2% in 2016. However, the proportion of women in the poorest and richest categories of wealth status remains almost similar throughout the study period.

### Poorest-richest gap in accessing safe childbirth care

Fig 1A shows the percentage of institutional delivery and Fig 1B percentage of births with skilled birth attendance in both poorest and richest categories of wealth status. They indicated

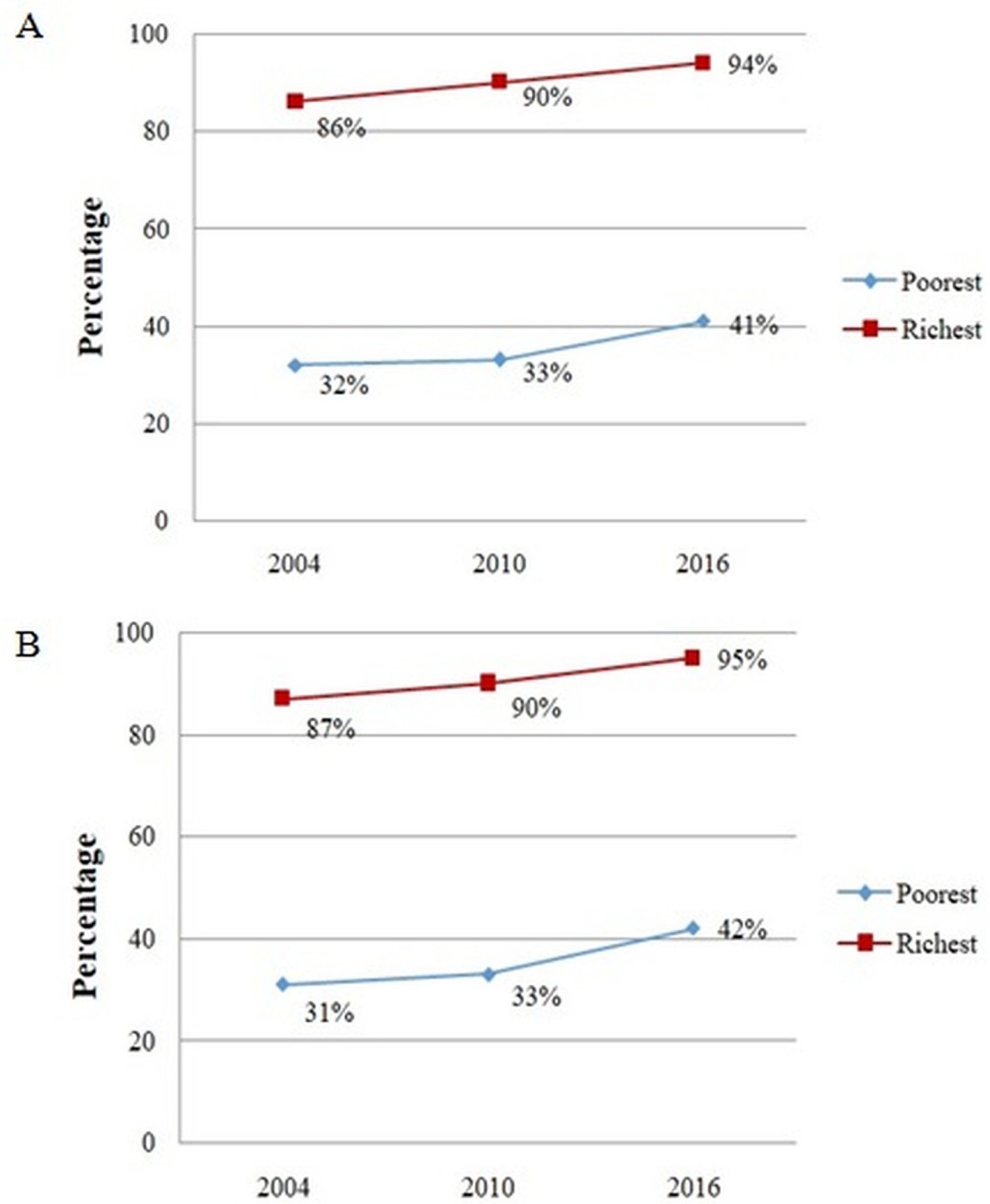

**Fig 1. Percentage distribution in accessing safe childbirth care between poorest and richest in Tanzania, 2004–2016.** (A) Difference in the percentage of between poorest and richest in accessing institutional delivery. (B) The difference in the percentage between poorest and richest in accessing skilled birth attendance.

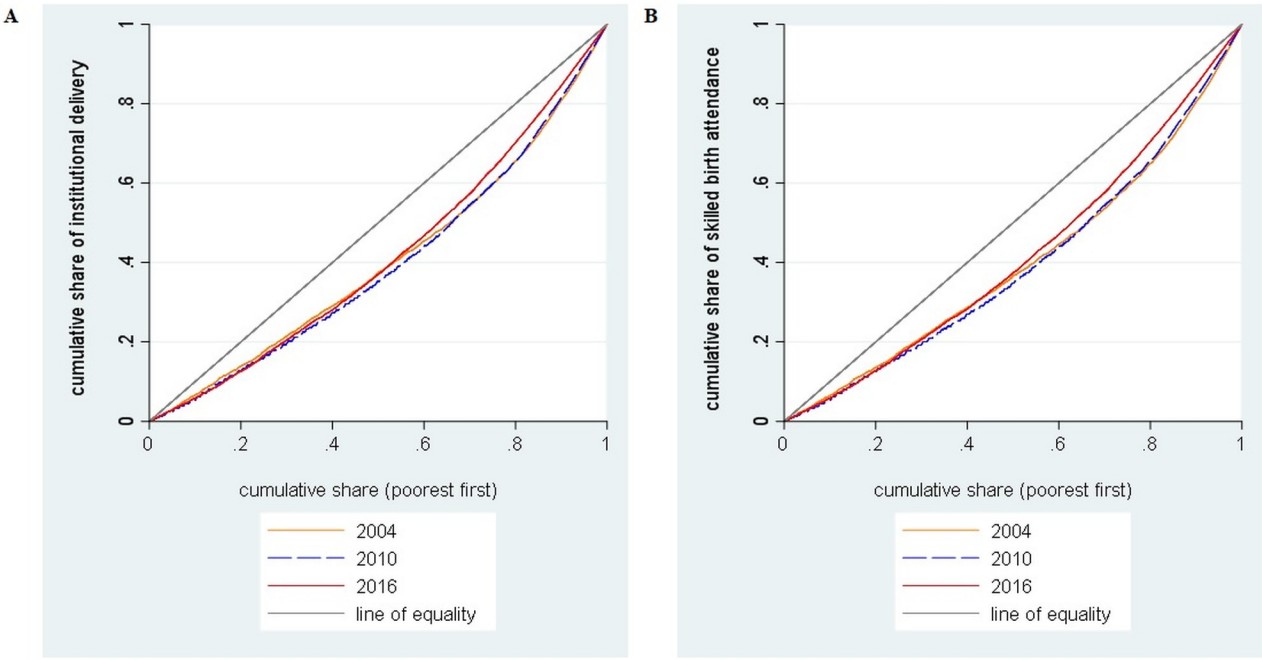

**Fig 2. Concentration curves in accessing safe childbirth care services in Tanzania DHS, 2004–2016.** (A) Concentration curves in accessing institutional delivery. (B) Concentration curves in accessing skilled birth attendance.

that there is a huge difference or gap (>50%) in accessing these two important indicators of safe childbirth care between the poorest and richest during the study period. (Note: the middle three categories of wealth were excluded only in this summary because of non-significant difference in percentages of accessing safe childbirth care).

## Patterns and trend of inequalities in accessing safe childbirth care

Fig 2A and 2B shows the concentration curves of institutional delivery and skilled birth attendance respectively for the year 2004, 2010, and 2016. The figures show that the curves were below the line of inequality which means inequalities in accessing institutional delivery and skilled birth attendance were disproportionately concentrated among the women from wealthier (rich) households. The nature of the curves suggested that the level of inequality did not decline over the years. These findings were confirmed with those presented in Table 2 which shows the magnitude of inequalities as indicated by the concentration indices. The results show that for all study years the concentration indices were positive which means there was pro-rich distribution in accessing institutional delivery and skilled birth attendance. Furthermore, there was no improvement in the reduction of inequalities over the past 12 years which reflects that the distribution remains highly pro-rich.

**Table 2. Concentration Indices for accessing safe childbirth care with household ranked by socioeconomic in Tanzania DHS, 2004–2016.**

|  | 2004 | 2010 | 2016 |
|---|---|---|---|
|  | Concentration Index [95% CI] | Concentration Index [95% CI] | Concentration Index [95% CI] |
| Institutional delivery | 0.195 [0.162–0.228] | 0.211 [0.184–0.237] | 0.175 [0.155–0.195] |
| Skilled birth attendance | 0.204 [0.173–0.234] | 0.213 [0.186–0.240] | 0.171 [0.152–0.190] |

Table 3. Contributing factors to the inequalities in accessing institution delivery in Tanzania DHS, 2004–2016.

| Variable | 2004 | | 2010 | | 2016 | |
|---|---|---|---|---|---|---|
| | Contribution | Percentage Contribution | Contribution | Percentage Contribution | Contribution | Percentage Contribution |
| **Maternal age** (ref. 15–19) | | | | | | |
| 20–24 | -0.00013773 | -0.07 | 0.00008686 | 0.04 | 0.00017952 | 0.10 |
| 25–29 | -0.00186078 | -0.95 | -0.00087544 | -0.42 | -0.00165373 | -0.95 |
| 30–34 | -0.00040123 | -0.21 | -0.00116987 | -0.56 | -0.00174259 | -1.00 |
| 35–39 | 0.00091594 | 0.47 | 0.00340514 | 1.62 | 0.00018299 | 0.10 |
| 40–44 | 0.00193968 | 0.99 | 0.00037526 | 0.18 | 0.00235923 | 1.35 |
| 45–49 | 0.00137132 | 0.70 | 0.00104978 | 0.50 | 0.00070268 | 0.40 |
| **Wealth index** (ref. Poorest) | | | | | | |
| Poor | -0.00579961 | -2.97 | -0.0035634 | -1.69 | -0.00870222 | -4.97 |
| Middle | 0.00160743 | 0.82 | 0.00532474 | 2.53 | 0.00489504 | 2.80 |
| Rich | 0.02826501 | 14.50 | 0.03996438 | 18.98 | 0.03899178 | 22.29 |
| Richest | 0.09062864 | 46.49 | 0.08469064 | 40.22 | 0.07745086 | 44.28 |
| Residence (ref. Rural) | | | | | | |
| Urban | 0.05168552 | 26.51 | 0.03880018 | 18.43 | 0.01717197 | 9.82 |
| **Maternal education** (ref. None) | | | | | | |
| Primary | 0.01017321 | 5.22 | 0.00682989 | 3.24 | -0.00098961 | -0.57 |
| Secondary | 0.00841181 | 4.31 | 0.01640126 | 7.79 | 0.02704991 | 15.46 |
| Higher | 0.00575481 | 2.95 | 0.0010946 | 0.52 | 0.00305603 | 1.75 |
| **Marital status** (ref. without a spouse) | | | | | | |
| Living with spouse | 0.00049764 | 0.26 | 0.00025359 | 0.12 | 0.0002194 | 0.13 |
| **Employment** (ref. Not employed) | | | | | | |
| Employed | 0.00074625 | 0.38 | 0.00772848 | 3.67 | 0.00696007 | 3.98 |
| **Region-effects** | 0.00029422 | 0.15 | -0.00090969 | -0.43 | 0.00413894 | 2.37 |
| Residual | 0.00086737 | 0.44 | 0.0110658 | 5.26 | 0.00465383 | 2.66 |

## Decomposition of concentration indices

Table 3 shows the decomposition of concentration indices of institutional deliveries per each study year. The results show that household wealth status was the largest contributor to inequality in accessing institutional delivery. It contributed to socio-economic inequalities for about 59%, 60%, and 74% in 2004, 2010, and 2016 respectively in favor of the privileged. The other largest contributor was the type of residence which contributed to about 27%, 18%, and 10% in 2004, 2010, and 2016 respectively in favor of the women living in urban. Maternal education increased inequality by about 13%, 12%, and 17% in 2004, 2010, and 2016 respectively in favor of women with at least a primary level of education.

Similarly, the decomposition results in Table 4 show that the household wealth status was the largest contributor to inequality in accessing skilled birth attendance followed by the type of residence and maternal education. Wealth status contributed to about 60%, 61%, and 65% in 2004, 2010, and 2016 respectively in favor of the women in the higher wealth status, while the place of residence contributed about 25%, 18%, and 9% in 2004, 2010, and 2016 respectively in favor of those living in the urban areas. Another significant contributor was maternal education, which contributed to the inequality by about 13%, 12%, and 17% in 2004, 2010, and 2016 respectively in favor of women with primary, secondary, higher education levels.

As the overall concentration indices for both outcome variable were positive in all study years, any contributing variables with a significant positive contribution (such as middle, rich

**Table 4. Contributing factors to the inequalities in accessing skilled birth attendance in Tanzania DHS, 2004–2016.**

| Variable | 2004 | | 2010 | | 2016 | |
|---|---|---|---|---|---|---|
| | Contribution | Percentage Contribution | Contribution | Percentage Contribution | Contribution | Percentage Contribution |
| **Maternal age** (ref. 15–19) | | | | | | |
| 20–24 | -0.00012934 | -0.06 | -0.00000555 | 0.00 | 0.0001917 | 0.11 |
| 25–29 | -0.00166078 | -0.82 | -0.00090915 | -0.43 | -0.00178453 | -1.04 |
| 30–34 | -0.00036906 | -0.18 | -0.00121641 | -0.57 | -0.00192369 | -1.13 |
| 35–39 | 0.00078619 | 0.39 | 0.00317104 | 1.49 | 0.00020028 | 0.12 |
| 40–44 | 0.00152849 | 0.75 | 0.00035026 | 0.16 | 0.00257212 | 1.50 |
| 45–49 | 0.00115693 | 0.57 | 0.00094557 | 0.44 | 0.0006493 | 0.38 |
| **Wealth index** (ref. Poorest) | | | | | | |
| Poor | -0.00595937 | -2.92 | -0.00322008 | -1.51 | -0.00872168 | -5.10 |
| Middle | 0.00154018 | 0.76 | 0.00600514 | 2.82 | 0.00489164 | 2.86 |
| Rich | 0.0299022 | 14.68 | 0.04162191 | 19.53 | 0.0386004 | 22.58 |
| Richest | 0.09673098 | 47.47 | 0.08627494 | 40.49 | 0.07692481 | 44.99 |
| **Residence** (ref. Rural) | | | | | | |
| Urban | 0.05162925 | 25.34 | 0.03760916 | 17.65 | 0.01498128 | 8.76 |
| **Maternal education** (ref. None) | | | | | | |
| Primary | 0.01108826 | 5.44 | 0.00667273 | 3.13 | -0.00093645 | -0.55 |
| Secondary | 0.00847612 | 4.16 | 0.01717725 | 8.06 | 0.02674101 | 15.64 |
| Higher | 0.005952 | 2.92 | 0.00104373 | 0.49 | 0.00282463 | 1.65 |
| **Marital status** (ref. without a spouse) | | | | | | |
| Living with spouse | 0.0006767 | 0.33 | 0.00025813 | 0.12 | 0.00012659 | 0.07 |
| **Employment** (ref. Not employed) | | | | | | |
| Employed | 0.00088284 | 0.43 | 0.00787692 | 3.70 | 0.00694415 | 4.06 |
| **Region-effects** | 0.00035886 | 0.18 | -0.00089657 | -0.42 | 0.00400733 | 2.34 |
| Residual | 0.00116285 | 0.57 | 0.01032388 | 4.85 | 0.00467861 | 2.74 |

and richest categories of wealth status, urban type of residence, secondary and higher education level) in Tables 3 and 4 means that inequalities in accessing births in health facilities and skilled birth attendance would have been less pro-rich if the concentration indices of these contributing variables were zero (i.e. were evenly distributed among the poor and the rich). The negative contributing variable (such as the poorer category of wealth status in Tables 3 and 4), would otherwise have increased the pro-rich distribution for accessing births in health facilities and skilled birth attendance if the concentration index of the contributing variable was zero.

## Discussion

This is the first study in Tanzania to describe the magnitude, time trend, and contributors of socioeconomic inequalities in accessing safe childbirth care based on two services which were births in health facilities (institutional delivery) and skilled birth attendance. The findings showed a persistent huge poorest–richest gap (> 50%) in accessing these two safe childbirth care services. This suggests that the effort of introducing exemption and waiver directives for user-fees to cover maternal health services in Tanzania did not eliminate differences between the poorest and richest in accessing safe childbirth care services. This might be due to the failure of local authorities to monitor its implementation, which resulted in it's ineffectively or inconsistently applied [24, 25]. Besides, the absence of a formal policy on waivers and

exemptions directives might contribute to the lack of legal weight for its effective implementation and leaving pregnant women especially in rural areas paying for delivery-related costs [26, 27]. The presence of any cost which could be affordable to richer than poorer women might result in this persistent poor–rich gap in accessing safe childbirth care services in Tanzania. Our findings corroborate with findings from other previous studies which reported a huge poor–rich gap in accessing maternal healthcare services [15, 28].

Furthermore, the findings from concentration indices and concentration curves highlighted the significant persistence of inequalities in accessing institutional delivery and skilled birth attendance from 2004 to 2016 among women in Tanzania. The estimates indicate that these inequalities were pro-rich meaning that women from wealthier households were privileged in accessing institutional delivery and skilled birth attendance compared to women from poorer households. These persistent inequalities indicate that the implementation of various maternal healthcare interventions such as increased coverage of health facilities that provide delivery services and the number of skilled birth attendants did not guarantee reduction or elimination of inequalities in accessing safe childbirth care services in Tanzania. However, on the other hand, these inequalities might be due to health system factors such as disproportionate availability of basic emergency obstetrics care (BemOC) [29], which is an important aspect for the facilities that provide normal deliveries to manage complications as they occur [30]. The previous report showed that less than 20% of dispensaries offered BemOC [31], and these are the most accessible facilities for women from poorer households particularly in rural areas [32]. Our findings are consistent with those from previous studies which show pro-rich socioeconomic inequalities in accessing safe childbirth care services in other SSA countries [21, 22].

Also, this study used a decomposition analysis to underscore the nature of observed persistent inequalities in accessing safe childbirth care services. The analysis was able to unveil that wealth status, place of residence, and maternal education as the major contributors to these persistent inequalities. It implies that these variables appear to prevent women from poorer households to access institutional delivery and skilled birth attendance services. Of these three major factors, wealth status was identified as the largest contributor to the inequalities in accessing these safe childbirth care services. If inequality in household wealth status were eliminated for example in 2016, it might have reduced the inequalities (by increasing the proportion of women from poorer households) in accessing institutional delivery and skilled birth attendance by 74% and 51% respectively. Therefore, eliminating wealth status inequality would be the first step towards reducing inequalities in accessing institutional delivery and skilled birth attendance. Reduction of this wealth status inequality might be achieved through a call for an integrated policy approach interventions which include fiscal policies, government spending, social protection, labor market, and employment policies, among others. The positive concentration indices and concentration curves below the line of equalities observed in this study is consistent with findings of other previous studies conducted elsewhere [19, 21, 33].

Also, the place of residence has been determined as another important contributor to the inequalities in accessing institutional delivery and skilled birth attendance services. It contributed to about 18% and 10% to inequalities in accessing institutional delivery and skilled birth attendance services in 2010 and 2016 respectively. This indicated that poor women from rural areas were less likely to access these two safe childbirth care services. The observed results might be due to the impact of high poverty levels at rural areas in Tanzania [34], in which most of the included respondents in this study were from those areas. However, much exploration is needed to underscore the background characteristics which subjected the poor women in rural areas to have less access to institutional delivery and skilled birth attendance services.

Regardless of inequalities, factors such as community perception towards health workers as well as realizing women´s right to maternal healthcare play significant roles in accessing safe childbirth care services especially in the rural areas of Tanzania [13, 35, 36]. Together with other factors such as obtaining permission, obtaining money to pay for consultation or treatment; distance to the facility, and absence of a companion to go to the facility were reported as barriers that needed to be addressed to improve the access of safe childbirth care services in Tanzania [16].

Furthermore, the decomposition analysis indicated significant emphasis on the contribution of maternal education in addressing inequalities in accessing institutional delivery and skilled birth attendance services. The results showed that if the elimination of inequality in women's education was achieved in 2016, it might have reduced the socioeconomic inequalities (by increasing the proportion of women from poorer households) in accessing both institutional delivery and skilled birth attendance by 17%. This is consistent with previous studies which highlighted maternal education as a major factor in women's decision to seek safe childbirth care services in Tanzania [37, 38]. Also, it has been shown that the majority of households with low socioeconomic status in Tanzania and other low-resource setting areas have low educational levels [16]. Therefore, achieving equalities in accessing institutional delivery and skilled birth attendance should also target education policies by considering the improvement of education for women from poorer households.

To the best of our knowledge, this is the first study of long-term trends of inequalities in accessing important safe childbirth care services in Tanzania, which used data from three consecutive nationally representative surveys. The use of the decomposition analysis helps to identify the major drivers of inequalities in accessing safe childbirth care services in Tanzania. However, this study has some limitations, as a cross-sectional study design was used, causality assumptions could not be made. Therefore, the results should be interpreted with caution. Also, there is a risk of recall bias which may have been introduced as a result of included women who had a live birth in the five years preceding the surveys. This might lead to either an over-or underestimation of the association between outcome and explanatory variables. Furthermore, missing some information in the dataset such as which facilities provide basic or comprehensive emergency obstetric and newborn care might have an impact on the results of accessing safe manhood care between poor and rich women.

## Conclusion

The results of this study provided evidence for the persistent huge poor-rich gap in favor of pro-rich situation in accessing institutional delivery and skilled birth attendance. The inequalities in household wealth status, place of residence, and maternal education contributed largely to the inequalities in accessing safe childbirth care services. The calls for an integrated policy approach which includes fiscal policies, social protection, labor market and employment policies need to improve education and wealth status for women from poor households. This might be the first step toward achieving universal maternal health coverage in Tanzania.

## Acknowledgments

I would like to acknowledge ICF International, Rockville, Maryland, USA, through the DHS program for giving me permission and access to the 2004, 2010, and 2016 TDHS datasets.

## Author Contributions

**Conceptualization:** Deogratius Bintabara.

**Data curation:** Deogratius Bintabara.

**Formal analysis:** Deogratius Bintabara.

**Methodology:** Deogratius Bintabara.

**Project administration:** Deogratius Bintabara.

**Resources:** Deogratius Bintabara.

**Software:** Deogratius Bintabara.

**Visualization:** Deogratius Bintabara.

**Writing – original draft:** Deogratius Bintabara.

**Writing – review & editing:** Deogratius Bintabara.

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
