## [Decision Letter · Decision Letter 0]

12 Nov 2020

PONE-D-20-17662

Addressing the huge poor–rich gap of inequalities in accessing safe childbirth care: a first-step to achieving the universal maternal health coverage in Tanzania

PLOS ONE

Dear Dr. Bintabara,

Thank you for submitting your manuscript to PLOS ONE. After careful consideration, we feel that it has merit but does not fully meet PLOS ONE’s publication criteria as it currently stands. Therefore, we invite you to submit a revised version of the manuscript that addresses the points raised during the review process.

We look forward to receiving your revised manuscript.

Kind regards,

Federica Angeli

Academic Editor

PLOS ONE

Journal Requirements:

Reviewers' comments:

Reviewer's Responses to Questions

**Comments to the Author**

1. Is the manuscript technically sound, and do the data support the conclusions?

Reviewer #1: Yes

Reviewer #2: Yes

2. Has the statistical analysis been performed appropriately and rigorously? 

Reviewer #1: Yes

Reviewer #2: Yes

3. Have the authors made all data underlying the findings in their manuscript fully available?

Reviewer #1: Yes

Reviewer #2: Yes

4. Is the manuscript presented in an intelligible fashion and written in standard English?

Reviewer #1: Yes

Reviewer #2: Yes

5. Review Comments to the Author

Reviewer #1: Although the manuscript is not unique in the fact that this topic has been researched globally and many of the methods used by other authors. However, given that this is the first time this particular approach has been used, employing the Tanzanian DHS, adds value to existing knowledge to what pertains to Tanzania. This work is publishable if the following suggestions are addressed.

1. Abstract and Introduction:

There are a number of typographical and grammatical errors in these sections and also in the discussion section. This makes it difficult to appreciate the information being provided. I would suggest major editing and correction to bring the manuscript to publishable level.

2. Methodology

For the data collection section, the authors need to provide references since this current version presumes that the study was carried out by the authors. Since they are referring to the data collection undertaken by a separate entity, the citations need to be appropriately placed to avoid any accusation of plagiarism. This pertains to the other sections where reference is made wither to the collection of the data or analysis performed such as the PCA.

3. Results

Were there other background variables that could have been added as contributory factors to explain the inequality in access to the key dependent variables? What about the effect of religious or cultural beliefs, which have been noted by some authors as key explanatory variables?

4. Discussion

The demographic results show some significant change in the rural-urban mix. For instance from 80% rural in 2004 to 72% in 2016. How does this particular change affect the contribution of this variable in the concentration index? We expect this trend to affect how these concentration curves perform as well.

5. Conclusion

The policy recommendations are quite generic and one would expect more specific ones in tandem to policy changes observed within the study period.

Reviewer #2: Thank you for the chance to review, “Addressing the huge poor-rich gap of inequalities in accessing safe Childbirth care: a first step to achieving the universal maternal health coverage in Tanzania.” I believe the analysis provides insight to deep inequalities in access to facility-based childbirth services and skilled birth attendance. The new way of analyzing the data is helpful (concentration curves, concentration indices, and decomposition analysis), but I think readers will need some guidance on how to understand them. A few points for consideration:

Major:

1. In Table 1 and the presentation of the DHS data, thee author notes the 5 categories of wealth; yet only thee poorest is compared to the richest. It would be helpful to understand why the middle three categories were excluded from analysis. Are low and middle income families more similar, with respect to access, to the poorest community or to the richest?

2. As noted above, the concentration index was new to me and may be new to readers. The methods presentation is clear, as are the results in the text. But it would be helpful to present more detail on the interpretation of the curves and the index (Table 2) together. Can the author provide more detail on how the two methods compare and contrast?

3. In maternal health work, the level of facility and/or provider may have a significant impact on health outcomes. In this analysis, BeMONC and CeMONC facilities are lumped together for “institutional delivery.” Did the authors examine the differences in access by facility type? How would this access impact results? In other words, are all the richest women receiving care at once type of facility? The authors should list this as a potential limitation.

4. Similarly, skilled birth attendant (SBA) definitions have been problematic. Did the authors examine the differences in access or inequality by SBA? Are all the richest women receiving care at one SBA type? The authors should list this as a potential limitation in terms of definition.

Minor

1. Although only one author is listed, the paper language uses “we” and “us.” Are there co-authors or people in acknowledgements?

2. Please be very clear in the abstract that richest women have higher access to the institutional delivery and skilled birth attendance. The current phrasing could be read as the poorest women or richest women have unequal access, but unclear who has unequal access.

3. In the data availability section, will the author provide analysis supporting materials (i.e. data analysis code, etc.)?

6. PLOS authors have the option to publish the peer review history of their article (what does this mean?). If published, this will include your full peer review and any attached files.

Reviewer #1: **Yes: **Ama Pokuaa Fenny

Reviewer #2: No

---

## [Author Response · Author response to Decision Letter 0]

12 Jan 2021

Response to reviewers

The author is very grateful for the reviews provided by the reviewers regarding this manuscript. I addressed all comments provided by the reviewers accordingly to make sure our article meet the PLOS ONE standards. 

Please see below, my detailed responses to the reviewers’ comments

Reviewer #1: 

General comments

Although the manuscript is not unique in the fact that this topic has been researched globally and many of the methods used by other authors. However, given that this is the first time this particular approach has been used, employing the Tanzanian DHS, adds value to existing knowledge to what pertains to Tanzania. This work is publishable if the following suggestions are addressed.

Authors’ response: Thank you for your comments and taking your time to review this manuscript.

Comment 1: Abstract and Introduction:

There are a number of typographical and grammatical errors in these sections and also in the discussion section. This makes it difficult to appreciate the information being provided. I would suggest major editing and correction to bring the manuscript to publishable level.

Response: Thank you for the comments. The typos and grammatical errors were edited and corrected throughout the document as suggested by the reviewer. 

Comment 2: Methodology

For the data collection section, the authors need to provide references since this current version presumes that the study was carried out by the authors. Since they are referring to the data collection undertaken by a separate entity, the citations need to be appropriately placed to avoid any accusation of plagiarism. This pertains to the other sections where reference is made wither to the collection of the data or analysis performed such as the PCA.

Response: Thank you for the comments. The reference were added as suggested by the reviewer; see [Reference [11] in Page 6, Line 10 and Page 11, Line 19]. 

Comment 3: Results

Were there other background variables that could have been added as contributory factors to explain the inequality in access to the key dependent variables? What about the effect of religious or cultural beliefs, which have been noted by some authors as key explanatory variables?

Response: Thank you for the comments. Since this is a secondary analysis, the dataset does not have such information. Therefore, we added this as one of the limitations of this study.

4. Discussion

The demographic results show some significant change in the rural-urban mix. For instance from 80% rural in 2004 to 72% in 2016. How does this particular change affect the contribution of this variable in the concentration index? We expect this trend to affect how these concentration curves perform as well.

Response: Thank you for the comments. The dataset from all three survey years were merged together and during analysis I adjusted for the year of surveys. Therefore, the obtained results have been adjusted for that factor (differences in year of surveys) 

Comment 5: Conclusion

The policy recommendations are quite generic and one would expect more specific ones in tandem to policy changes observed within the study period.

Response: Thank you for the comments. Based on the findings from this study, it seems the integrated policy approach is needed to makes sure women from poor households have similar access to childbirth care as those from wealthier households. And this is what I recommended see [Page 3, Line 5 – 6 and Page 24, Line 2 – 4].

Reviewer #2: 

Thank you for the chance to review, “Addressing the huge poor-rich gap of inequalities in accessing safe Childbirth care: a first step to achieving the universal maternal health coverage in Tanzania.” I believe the analysis provides insight to deep inequalities in access to facility-based childbirth services and skilled birth attendance. The new way of analyzing the data is helpful (concentration curves, concentration indices, and decomposition analysis), but I think readers will need some guidance on how to understand them. A few points for consideration:

Response: Thank you for the comments and taking your time to review this manuscript.

Major:

Comment 1: In Table 1 and the presentation of the DHS data, thee author notes the 5 categories of wealth; yet only thee poorest is compared to the richest. It would be helpful to understand why the middle three categories were excluded from analysis. Are low and middle income families more similar, with respect to access, to the poorest community or to the richest?

Response: Thank you for the comments. The middle three categories were not excluded from the analysis except only for summarizing the gap between poorest and richest. To avoid misunderstanding the details has been added in Result section; see [Page 13, Line 15 – 17]. 

Comment 2: As noted above, the concentration index was new to me and may be new to readers. The methods presentation is clear, as are the results in the text. But it would be helpful to present more detail on the interpretation of the curves and the index (Table 2) together. Can the author provide more detail on how the two methods compare and contrast?

Response: Thank you for the comments. The details on the interpretation of curves and index have been clearly presented on [Page 9, Line 13 – 22 to Page 10 Line 6 – 13]. 

Comment 3: In maternal health work, the level of facility and/or provider may have a significant impact on health outcomes. In this analysis, BeMONC and CeMONC facilities are lumped together for “institutional delivery.” Did the authors examine the differences in access by facility type? How would this access impact results? In other words, are all the richest women receiving care at once type of facility? The authors should list this as a potential limitation.

Response: Thank you for the comments. This was added as one of the limitation as suggested by the reviewer.

Comment 4: Similarly, skilled birth attendant (SBA) definitions have been problematic. Did the authors examine the differences in access or inequality by SBA? Are all the richest women receiving care at one SBA type? The authors should list this as a potential limitation in terms of definition.

Response: Thank you for the comments. Assisted by any of the personnel listed in the definition regardless of the wealth status categories was considered as skilled birth attendance; see [Page 8, Line 1 – 3].

Minor:

Comment 1: Although only one author is listed, the paper language uses “we” and “us.” Are there co-authors or people in acknowledgements?

Response: Thank you for the comments. There is only one author for this work. The typos have been checked and amended throughout the document. 

Comment 2: Please be very clear in the abstract that richest women have higher access to the institutional delivery and skilled birth attendance. The current phrasing could be read as the poorest women or richest women have unequal access, but unclear who has unequal access.

Response: The sentence has been revised and amended for clear understand; see [Page 2, Line 19 – 21]. 

Comment 3: In the data availability section, will the author provide analysis supporting materials (i.e. data analysis code, etc.)?

Response: Thank you for the comments. The data used for analysis is free available from DHs program website the codes will be shared upon request to the author (email: bintabaradeo@gmail.com).

---

## [Decision Letter · Decision Letter 1]

1 Feb 2021

Addressing the huge poor–rich gap of inequalities in accessing safe childbirth care: a first step to achieving universal maternal health coverage in Tanzania

PONE-D-20-17662R1

Dear Dr. Bintabara,

We’re pleased to inform you that your manuscript has been judged scientifically suitable for publication and will be formally accepted for publication once it meets all outstanding technical requirements.

Kind regards,

Federica Angeli

Academic Editor

PLOS ONE

Additional Editor Comments (optional):

Reviewers' comments:

Reviewer's Responses to Questions

**Comments to the Author**

1. If the authors have adequately addressed your comments raised in a previous round of review and you feel that this manuscript is now acceptable for publication, you may indicate that here to bypass the “Comments to the Author” section, enter your conflict of interest statement in the “Confidential to Editor” section, and submit your "Accept" recommendation.

Reviewer #1: All comments have been addressed

Reviewer #2: All comments have been addressed

2. Is the manuscript technically sound, and do the data support the conclusions?

Reviewer #1: Yes

Reviewer #2: Yes

3. Has the statistical analysis been performed appropriately and rigorously? 

Reviewer #1: Yes

Reviewer #2: Yes

4. Have the authors made all data underlying the findings in their manuscript fully available?

Reviewer #1: Yes

Reviewer #2: Yes

5. Is the manuscript presented in an intelligible fashion and written in standard English?

Reviewer #1: Yes

Reviewer #2: Yes

6. Review Comments to the Author

Reviewer #1: All comments I posed have been addressed by the author. The limitations section has been updated and grammatical and typographical errors corrected.

Reviewer #2: Thank you for responding to my original review comments as well as the other reviewer. I have no additional comments at this time.

7. PLOS authors have the option to publish the peer review history of their article (what does this mean?). If published, this will include your full peer review and any attached files.

Reviewer #1: **Yes: **Ama Fenny

Reviewer #2: No

---

## [Editor Report · Acceptance letter]

4 Feb 2021

PONE-D-20-17662R1 

Addressing the huge poor–rich gap of inequalities in accessing safe childbirth care: a first step to achieving universal maternal health coverage in Tanzania 

Dear Dr. Bintabara:

I'm pleased to inform you that your manuscript has been deemed suitable for publication in PLOS ONE. Congratulations! Your manuscript is now with our production department. 

Kind regards, 

on behalf of

Prof. Federica Angeli 

Academic Editor

PLOS ONE